# Differential colitis susceptibility of Th1- and Th2-biased mice: A multi-omics approach

**Sohini Mukhopadhyay**[1,2], **Subha Saha**[3], **Subhayan Chakraborty**[2,4], **Punit Prasad**[3], **Arindam Ghosh**[iD][2,4], **Palok Aich**[iD][1,2]*

1 School of Biological Sciences, National Institute of Science Education and Research (NISER), HBNI, Khurdha, Odisha, India, 2 Homi Bhabha National Institute, Training School Complex, Anushaktinagar, Mumbai, India, 3 Institute of Life Sciences, NALCO Square, Bhubaneswar, Odisha, India, 4 School of Chemical Sciences, National Institute of Science Education and Research (NISER), HBNI, Khurdha, Odisha, India

* palok.aich@niser.ac.in

**Data Availability Statement:** All relevant data are within the paper and its Supporting information files.

## Abstract

The health and economic burden of colitis is increasing globally. Understanding the role of host genetics and metagenomics is essential to establish the molecular basis of colitis pathogenesis. In the present study, we have used a common composite dose of DSS to compare the differential disease severity response in C57BL/6 (Th1 biased) and BALB/c (Th2 biased) mice with zero mortality rates. We employed multi-omics approaches and developed a newer vector analysis approach to understand the molecular basis of the disease pathogenesis. In the current report, comparative transcriptomics, metabonomics, and metagenomics analyses revealed that the Th1 background of C57BL/6 induced intense inflammatory responses throughout the treatment period. On the contrary, the Th2 background of BALB/c resisted severe inflammatory responses by modulating the host's inflammatory, metabolic, and gut microbial profile. The multi-omics approach also helped us discover some unique metabolic and microbial markers associated with the disease severity. These biomarkers could be used in diagnostics.

## Introduction

Inflammatory bowel diseases (IBD), consisting of Crohn's disease and ulcerative colitis, are an increasing global threat. During the 20th century, IBD was mainly a disease of westernized countries. At the turn of the 21st century, IBD became a global disease with accelerating incidence in the newly industrialized countries of Asia, South America, and Africa. The incidence of IBD in Asia is 1.4 cases per 100,000, just after the USA, with a prevalence of 1.6 cases per 100,000. In Asia, ulcerative colitis is 2-fold more likely to be diagnosed than Crohn's disease than the other parts of the world. The highest incidence of colitis in Asia–Pacific is in India at 9.3 per 100,000 persons/years [1–5].

Although the condition is alarming, no unique cause has been determined for colitis. Still, its etiopathogenesis is thought to arise from genetic susceptibility to dysregulated interaction between immune factors and the enteric commensal flora. Environmental triggers, such as

**Funding:** The funders had no role in study design, data collection and analysis, decision to publish, or preparation of the manuscript.

**Competing interests:** The authors have declared that no competing interests exist.

drug use, stress, diet, and smoking, also influence disease onset and development [6]. Research progress to date in this area indicates that colitis is a multifactorial disease. To understand the heterogeneous outcome of the disease, a systems biology approach aiming to integrate biological omics and non-omics datasets can be a solution to resolve the complexity of the disease. The multi-omics approach is the best way to provide vast information about the therapeutic strategy and discover clinical biomarkers that characterize disease pathogenesis.

We vigorously followed the multi-omics approach to understand colitis's disease onset and etiology in the current study. To avoid ethical issues related to human samples, we have used the mouse as a model system to study disease pathogenesis. Typically, in a colitis study, the disease is induced using chemical compounds, e.g., DSS, or by knocking out or targeting specific genes, such as regulatory cytokines. However, the DSS-induced colitis mouse model is prevailing because of its similarities with human colitis and rapidity, simplicity, reproducibility, and controllability [7, 8]. Different DSS dosages led to the development of robust colitis models in mice [9]. Studies reported that the optimum DSS dose varied with varying strains of mice. The optimal concentration recommended for inducing colitis was 1.5%- 3.0% in C57BL/6 (Th1 bias) and 2.5–5.0% in BALB/c (Th2 bias) mice [8, 10–12]. There is no standard optimized DSS dose available with zero mortality rates that can cause remarkable disease outcomes for the same duration independent of mice strains or immunological background. We established a common composite dose of DSS in the current study, i.e., 5% DSS for 7 days followed by 2.5% DSS for another 7 days for C57BL/6 and BALB/c mice.

The current report revealed that the Th1 background of C57BL/6 mice might be responsible for flares of inflammation throughout the treatment period irrespective of DSS concentration. Altered inflammatory responses disrupt the homeostasis of the host in such a way that they transform the metabolism and gut microbial composition of the host to create a more intense form of colitis. On the contrary, the Th2 background of BALB/c was unable to activate severe inflammatory reactions throughout the treatment condition. The severity response for the initial treatment condition was probably the outcome of the altered gut microbial composition of the host rather than the host genetic response. Lowering of DSS concentration caused recovery of the host from the diseased state. The current multi-omics study, along with the vector analysis, revealed that a) higher DSS concentration could not make many changes in genes and host metabolic composition, but b) the gut microbial composition and microbial metabolites created a niche that supports inflammation at the initial days of the disease. The host regained normal homeostasis and activated more anti-inflammatory responses as the gut microbial composition and metabolic diversity was restored at lower DSS concentration.

We further established a few probable metabolomic and gut microbial markers independent of host immunological conditions. We observed that inflammation leads to the host system's high amino acid and lipid metabolism, leading to the high abundance of Helicobacter genus under Proteobacteria phylum in the gut. These parameters all together create a niche for the severe form of colitis in the host. These biomarkers would help us understand the disease etiology better and could help us diagnose the disease at its early onset.

## Materials and methods

### Mice

We co-housed specific pathogen-free 6–8 weeks old male C57BL/6-or BALB/c-mice with body weight in the range of 18-22g in a poly-sulfone cage using corncob as bedding material. We housed the mice of the same strains together in a pathogen-free environment with a 12h light and 12h dark cycle at 24 ± 3˚ in 55 ± 3% humidity. We provided traditional pelleted food and autoclaved water ad libitum. Following one week of the acclimatization period, we randomly

grouped the animals into 3 groups and 6 animals, based on their strain for further experiments. Committee for Control and Supervision of Experiments on Animals, Govt. of India (CPCSEA) approved this study. We performed all experiments as per the approved guidelines.

## Colitis induction and sample collection

To induce colitis in both the strains of mice, we added DSS (M.W.-50kDa, Fisher Scientific) to autoclaved drinking water (*ad libitum*) at a concentration of 5% for the 1st week, followed by 2.5% of DSS for the 2nd week, and renewed with freshly prepared solution three times a week. This particular dose was able to induce significant colitis in both strains with zero mortality rate. We fed the Control group with normal autoclaved water.

Among the 3 groups based on mice strains, 2 groups consumed DSS water and were marked as treated groups. The other group consumed normal drinking water and kept it as control. We sacrificed mice on days 0, 7, and 15 and collected samples from the cecum and colon. Euthanasia was performed by introducing 100% carbon dioxide into a bedding-free cage initially containing room air with the lid closed at a rate sufficient to induce rapid anesthesia, with death occurring within 2.5 minutes. Mice were sacrificed if they reached the humane endpoints of rectal prolapse, loss of >15% body weight, or signs of pain and distress including poor grooming, decreased activity, and hunched posture to alleviate the suffering of the animals.

We snap-frozen the colon tissue samples for protein analysis and cecum contents for microbiome and metabolome-related studies and stored them at -80˚C. We collected colon tissue samples in RNALater for further RNA-related analysis. We collected the whole blood by cardiac puncture for serum isolation. Serum was isolated by centrifuging the whole blood at 1600 rcf at 4˚C for 10 minutes and stored at -20˚C until further analysis.

## RNA extraction and lithium chloride purification

According to the manufacturer's protocol, we. We purified the samples further with LiCl to get rid of all polysaccharides, including DSS (a) by incubating RNA on ice with 0.1 volume of 8 M LiCl for 2 h followed by centrifuging at 14,000 g for 30 min at 4˚C, and (b) discarded the supernatant and dissolved RNA pellets in 200 μl of nuclease-free water. (c) We repeated steps (a-b) once more before precipitating RNA at −20˚C for 30 min, in 0.1extracted the total RNA from colon tissues using Qiagen RNeasy mini kit (Cat# 74104, Qiagen, India) volume of 3 M sodium acetate (pH 5.2) and 2 volumes of 100% absolute ethanol. (d) We centrifuged the RNA at 14,000 g for 30 min at 4˚C and discarded the supernatant. (e) Washed the pellets with 100 μl of 70% ethanol, centrifuged at 14,000 g for 10 min at 4˚C, and dissolved the final RNA in 30μl nuclease-free water. We determined the RNA quality and quantity by measuring ODs at 230, 260, and 280nm using NanoDrop 2000 machine (Thermo Fisher Scientific, Columbus, OH, USA) and validated it using a Qubit4 fluorometer (Invitrogen, California, USA). We assessed the RNA integrity by 2% agarose gel electrophoresis and confirmed by 4200 TapeStation instruments (Agilent, Santa Clara, CA, USA) [13].

## Total gene expression by mRNA sequencing

We submitted the samples to Agrigenome with RIN value above 8 for library preparation. Agrigenome prepared the library using TruSeq Stranded mRNA Library Prep Kit (poly-A selection) and sequenced it using the HiSeq-2500 platform (Illumina, San Diego, CA, USA). They sequenced the samples using pair-end 2*150-bp sequencing, aiming coverage of 20 M reads and used the Tuxedo software package for the pipeline for data analysis. The package consists of spliced read mappers and tools that allow one to assemble transcripts, estimate

their abundances, and test differential expression and regulation in RNA-Seq samples. It was a combination of open-source software and implemented peer-reviewed statistical methods. The components of the NGS data analysis pipeline for RNA-seq include Hisat2, StringTie, and Cuffdiff. We took data with a Phred score >30, alignment percentage >60. Replicate variability was significantly less and clustered according to the experimental group. Similar numbers of genes were expressed in all samples, and sort differentially expressed genes using relevant criteria, including expression level, fold change, and statistical significance [14, 15]. We used Statistixl 2.0 add-on to MS Excel for Linear Discriminant Analysis (LDA or DCA).

## cDNA Preparation

Following the manufacturer's protocol, we converted the mRNA into cDNA from total RNA using AffinityScript One-Step RT-PCR Kit (Cat# 600559, Agilent, Santa Clara, CA, USA).

## Quantitative Real-Time PCR (qRT PCR)

We set the qRT-PCR reaction in a 96 well PCR plate using a) 30 ng of cDNA as a template in the presence of 1μM/μl of each of forward (F) and reverse (R) primers (S1 Table) for genes mentioned in S1 Table, b) SYBR green master mix (Cat#A6002, Promega, Madison, WI, USA), and c) nuclease-free water. We performed the qRT-PCR analysis in Quantstudio 7 Real-Time PCR (Thermo Fisher Scientific, Columbus, OH, USA). We normalized the measured cycle threshold ($C_t$) value of a gene with the GAPDH (positive control) $C_t$-value, and the fold change of the desired gene was calculated to the control $C_t$-value.

## Intestinal permeability assay using FITC-dextran

We food and water-starved both control and treated mice of both strains overnight and kept them in a cage without bedding to limit the coprophagic behavior [16]. We measured mice body weights the following morning. We administered PBS dissolved FITC-dextran (100mg/ml, Cat#F7250, Sigma-Aldrich, Missouri, US) in each mouse (44mg/100g body weight) by oral gavaging following 4h of incubation. We anesthetized mice by isoflurane inhalation and collected blood using a 1ml syringe with a 25G needle by cardiac puncture. We isolated serum from the blood using the protocol described previously and diluted it with an equal volume of PBS. We added 100μl diluted serum in each well of a 96 well microplate in duplicate. We measured fluorescence at 528 nm (emission wavelength) by exciting at 485 nm (20 nm bandwidth at both excitation and emission) [17].

## C-Reactive Protein (CRP) assay

We collected the colon tissues from the control and treated group of mice and rapidly homogenized them in tissue in an extraction buffer (Cat#78510, Thermo Scientific, Rockford, USA) containing a 1X protease inhibitor cocktail (Cat#78429, Thermo Fisher Scientific, Rockford, USA). We centrifuged tissue homogenates at 13,000 rcf for 10 minutes at 4°C to remove the insoluble debris. Protein concentration was determined using Bradford assay (Cat#5000006, Bio-Rad, USA). We performed the CRP assay per the manufacturer's protocol (Cat# ELM-CRP, Norcross, GA).

## Sample preparation for serum metabolomics study

We used the freshly isolated serum, collected from the control and treated group, for this study and processed the serum samples following the previously described protocol by Naik *et al.* and Ray *et al.* [18, 19].

## Sample preparation for cecal metabolomics study

We used the cecal content stored at -80˚C for this study and processed the samples following the previously described protocol by Wu *et al.* [20].

## $^1$H NMR data acquisition and metabolite analysis

We performed all NMR experiments at 298K on a Bruker 9.4T (400 MHz) Avance-III Nano-bay solution-state NMR spectrometer equipped with a 5 mm broadband probe. We used excitation sculpting gradients with a duration of 1 ms and a strength of 14.9 G/cm for water suppression and an offset optimization using real-time 'gs' mode for each sample. In addition, we employed a Sinc-shaped pulse of 2 ms for selective excitation of the water resonance. We recorded 64 transients for each set of experiments with an average 5s relaxation delay to ensure complete water saturation. We recorded and processed the acquired spectra by Topspin 2.1.

We used Chenomx NMR Suite7.6 (ChenomxInc., Edmonton, Canada) to identify (targeted) and quantify metabolite signals from NMR spectra. First, the Chenomx processor automatically phased, referenced the DSS peak at 0 ppm, and corrected FID files' spectra' baseline. Next, we calculated the metabolite concentrations by a profiler using Metaboanalyst 5.0 (a public software at https://www.metaboanalyst.ca). We utilize the profiler to assign and fit the metabolite peaks from the Chenomax library. We performed the pathway analysis using Metaboanalyst 5.0 software.

We identified the statistically significantly altered metabolites between two experimental conditions from the serum and cecal metabolite concentration list. Then we further shortlisted the metabolites by setting up a cut-off of 1.5 based on the relative concentration of metabolites. Next, we clustered (based on all the metabolites and significantly altered metabolites) the data based on different treatment conditions using Linear Discriminant Analysis using statistiXL 2.0 (add-in to the Windows version of Excel spreadsheet).

Finally, we used the list of significantly altered metabolites to predict biological pathways using the MetPA function of the MetaboAnalyst 5.0 software. We used KEGG pathways as reference data set for biological pathway prediction. Briefly, we manually entered the compound lists into the pathway analysis module along with the necessary metadata. The metabolite lists generated an overview plot of the predicted pathways. Similarly to this, we have done joint pathway analysis, where software can predict the affected biological pathways based on metabolite level changes and genetic level changes. We manually entered significantly altered metabolites and significantly altered genes available from the transcriptomics study and its proper metadata to know the collective impact of host metabolism and host genetics on the overall biological system of the host [18, 21, 22].

From the list of predicted pathways, we chose the ones with high impact scores to be altered due to different treatment conditions, giving rise to the differential disease responses based on the differential immune bias condition of the host.

## Detailed analysis of transcriptomics and metabonomics data from LDA clustering

We performed Linear Discriminant Analysis (LDA) with the shortlisted genes, metabolites, and meta-metabolites to confirm the differential disease responses at various treatment conditions. We calculated the distance (r) between clusters, i.e., different treatment conditions on the 2D plane, to determine the differentiability of the changes of genes and metabolites based on treatment conditions and different mice strains. Distance between other clusters helped us quantify the dissimilarity in disease progression at different treatment conditions, and we

named it Dissimilarity Coefficient. We also drew the trajectory followed by different treatment conditions on the 2D plane to know the characteristics changes of genes and metabolites based on disease severity (S2 Table) [23, 24].

$$\text{Dissimilarity Coefficient } (\mathbf{r}) = \sqrt{(\boldsymbol{x_2} - \boldsymbol{x_1})^2 + (\boldsymbol{y_2} - \boldsymbol{y_1})^2}$$

Where,

$\mathbf{r}$ = distance between the point $x_1y_1$ and $x_2y_2$ ($x$, $y$ depicts the different treatment conditions).

To know the role of genes and metabolites in determining the disease severity based on host immunological background, we calculated the Disease Severity Index (D) considering three different parameters separately, i.e., altered genes and metabolites meta-metabolites expressions [25].

$$\text{Disease Severity Index } (\mathbf{D}) = \Sigma \boldsymbol{\rho_x \rho_y r_{xy}}$$

Where,

$\boldsymbol{\rho_x}$ = distance of point x from the center.

$\boldsymbol{\rho_y}$ = distance of point 'y' from the center.

$\boldsymbol{r_{xy}}$ = distance between points '$x$' and '$y$'.

$\Sigma$ = summation of $\boldsymbol{\rho_y}$, $\boldsymbol{r_{xy}}$ between two specific treatment conditions.

($x$, $y$ represents different treatment conditions).

We further checked the ratio of pro- and anti-inflammation-related genes, metabolites, and meta-metabolites at a particular treatment condition to understand the possible factors playing the leading role in activating the host's inflammatory conditions (S3–S5 Tables). We have depicted the Venn diagram of shortlisted genes of C57BL/6 and BALB/c to find the common and unique genes between two strains at various treatment conditions. The strain-specific ratio of pro- and anti-inflammatory genes gave us a clear idea about the reason behind differential disease severity in two different mice strains. Similarly, we have depicted the strain-specific Venn diagram of shortlisted metabolites from serum and cecal content to find the common and unique metabolites from serum and cecal content at various severity levels. The ratio of pro and anti-inflammatory metabolites from serum and cecal content helps us clear the doubt about the role of metabolites and meta-metabolites in host-specific inflammatory conditions. We plotted the number of significantly altered metabolites at different treatment conditions to know whether the diseased condition was responsible for altered serum and cecal content diversity. We also represented the Venn diagram of significantly altered common and unique serum and cecal metabolites of different severity levels. This analysis helped us find the similarities and dissimilarities between the differential composition of serum and cecal metabolites at various diseased conditions, which further correlated with the disease severity.

## Cecal DNA extraction and lithium chloride purification

We collected cecal samples from the control and treated group of the mice strains and stored them at −80 ˚C. As per the manufacturer's protocol, we isolated the genomic DNA from caecal content using QIAamp® DNA stool Minikit (Qiagen, Germany, Cat. No.# 51304). We purified the DNA samples further with LiCl to get rid of all polysaccharides, including DSS. Briefly, a) We incubated DNA on ice with 0.1 volume of 8 M LiCl for 2 h followed by centrifugation at 14,000 g for 30 min at 4˚C, and (b) discarded the supernatant and dissolved DNA pellets in 200 µl of nuclease-free water. (c) We repeated steps (a-b) once more before precipitating DNA

at −20˚C for 30 min, in 0.1 volume of 3 M sodium acetate (pH 5.2) and 2 volumes of 100% absolute ethanol. (d) We centrifuged the DNA at 14,000 g for 30 min at 4˚C and discarded the supernatant. (e) Washed the pellets with 100 μl of 70% ethanol, centrifuged at 14,000 g for 10 min at 4˚C, and dissolved the final DNA in 50μl nuclease-free water. We determined the quality and quantity of extracted DNA by measuring ODs at 230, 260, and 280nm using NanoDrop 2000 machine (Thermo Fisher Scientific, Columbus, OH, USA) and validated it using a Qubit4 fluorometer (Invitrogen, California, USA). We assessed the DNA integrity by 2% agarose gel electrophoresis and confirmed it by 4200 TapeStation instruments (Agilent, Santa Clara, CA, USA) [26, 27].

## 16S rRNA sequencing (V3-V4 metagenomics)

We amplified V3-V4 regions of the 16S rRNA gene of cecal DNA samples. For this amplification, we have used V3F: 5'-CCTACGGGNBGCASCAG-3' and V4R: 5'- GACTACNVGGGT ATCTAATCC-3' primer pair [28]. In the Illumina Miseq platform, amplicons were sequenced paired-end (250bpX2) with a sequencing depth of 500823.1 ± 117098 reads. We monitored the base composition, quality, and GC of the fastq sequence. More than 90% of the sequences had Phred quality score above 30 and GC content nearly 40–60%. We removed conserved regions from paired-end reads. We constructed a consensus V3-V4 region sequence using the FLASH program by removing unwanted sequences [29, 30]. We pre-processed reads from all the samples pooled and clustered into Operational Taxonomic Units (OTUs) using the de novo clustering method based on their sequence similarity using the UCLUST program. We used QIIME for the OTU generation and taxonomic mapping [31, 32]. Finally, we identified a representative sequence for each OTU and aligned it against the SILVA core set of sequencethes using the PyNAST program. Alignment of these representative sequences against reference chimeric data sets and RDP classifier against SILVA OTUs database was used for taxonomic classification [32–35].

## Microbiota composition profiling and analysis

We further analyzed the biome file with the phylogenetic information of the OTUs and filtered OTUs for minimum counts of 2 and a prevalence of 20%. We scaled the data using the total sum scaling algorithm. We plotted the stacked bar and line plot of the phylum and genus level classification.

## Microbial and metabolite evenness index

To get an idea about the altered microbial and metabolite diversity due to diseased conditions, we calculated the Evenness index value (E) of microbiota at the phylum level and the metabolites from serum and cecal content [19].

$$\text{Evenness Index (E)} = -\Sigma E_i ln E_i$$

Where,

$E_i$ = proportion of the individual phylum/ metabolites in the total microbial/metabolite pool.

$lnE_i$ = natural logarithm of $E_i$.

$-\Sigma$ = negative sum of $E_i lnE_i$ for an individual in a specific treatment condition.

## Results

We performed this study because colitis causes severe inflammatory changes in the host. Activation of the pro-inflammatory condition further leads to systemic level changes in the host, e.g., i) changes in gut microbial composition, ii) alteration of serum and cecal metabolite concentration, iii) changes in gut barrier functions, and acute phase responses of the host, etc. However, the role of host genetics, mainly the immunological background of the host, in colitis related complications remains very poorly understood to date. In this study, we attempted to address the role of host genetics, more specific, the role of host immunological background in case of i) extent of inflammatory changes of the host, ii) differential changes of gut microbial composition based on host genetic background, and severity of the disease, iii) alteration of metabolite profile of host-based on host genetics, disease severity, altered gut microbial composition too.

### Colitis induction and associated inflammatory changes in the host

We administered various DSS dosages orally in C57BL/6- and BALB/c- male mice. We used untreated mice as a control in the study. We tried several doses to determine an optimum DSS amount that would not cause mortality in either of the mice strains and found 5% and 2.5% for 2 weeks as optimum for both the strains. Oral administration of DSS (5% for the 1st week +2.5% for the 2nd week) for 2 weeks induced colitis related typical and notable physiological changes (e.g., loss of body weight, diarrhea, rectal bleeding, and presence of fecal occult blood- we have used these parameters to determine the disease severity/ clinical score) in both mice strains (S1A Fig). Along with the physiological alterations of the host, we measured inflammatory responses from the colon. Altered bowel/colon length and histopathological scoring were good enough to provide a distinct idea about the kinetics of disease progression in both the mice strains (S1B and S1C Fig). Data in the S1 Fig indicated that the disease progression continued until the treatment period in C57BL/6. BALB/c entered into the recovery phase after day 7 of the DSS treatment.

A summary slide mentioning the critical findings of the present model system is described in the S2 Fig.

As the first-line defense of host inflammatory responses, we measured the transcription level expression of Toll-Like Receptors (TLR) from mice's colonic tissue. TLRs can recognize damage-associated molecular patterns released from damaged tissues and play a vital role in activating the other pro-inflammatory responses of the host [36, 37]. We measured *Tlr2* and *Tlr4* expression of mouse colonic tissue and found that both the gene expressions were significantly higher in C57BL/6 mice than the BALB/c mice throughout the treatment condition. More surprisingly, we also found that *Tlr2* and *Tlr4* expressions continuously increased till the end of the treatment period. On the contrary, in BALB/c, both gene expressions were significantly increased until day 7, reaching their basal level on day 15 of the treatment (Fig 1A).

Excessive TLR activation disrupts the immune homeostasis by sustained pro-inflammatory cytokines and chemokine production [38, 39]. To investigate the effect of high TLR responses in the diseased condition, we analyzed transcriptional profiling of selected cytokines, i.e., *TNFα, IFNγ, IL1β, IL6, IL12, IL17, IL21, IL10*. The chosen cytokines played a crucial role in the disease progression. At the higher DSS dose (5%), gene expressions of all inflammatory markers, i.e., *TNFα, IFNγ, IL1β, IL6, IL12, IL21*, and *IL17*, were significantly higher in the treated group of mice compared to the control in both the strains (Fig 1B and 1C). In the presence of 2.5% DSS during the 2nd-week, pro-inflammatory cytokine expressions gradually increased in treated C57BL/6 mice till day 15 (Fig 1B and 1C). We did not observe any significant upregulation in the pro-inflammatory cytokine levels in DSS-treated BALB/c mice on the 2nd week of

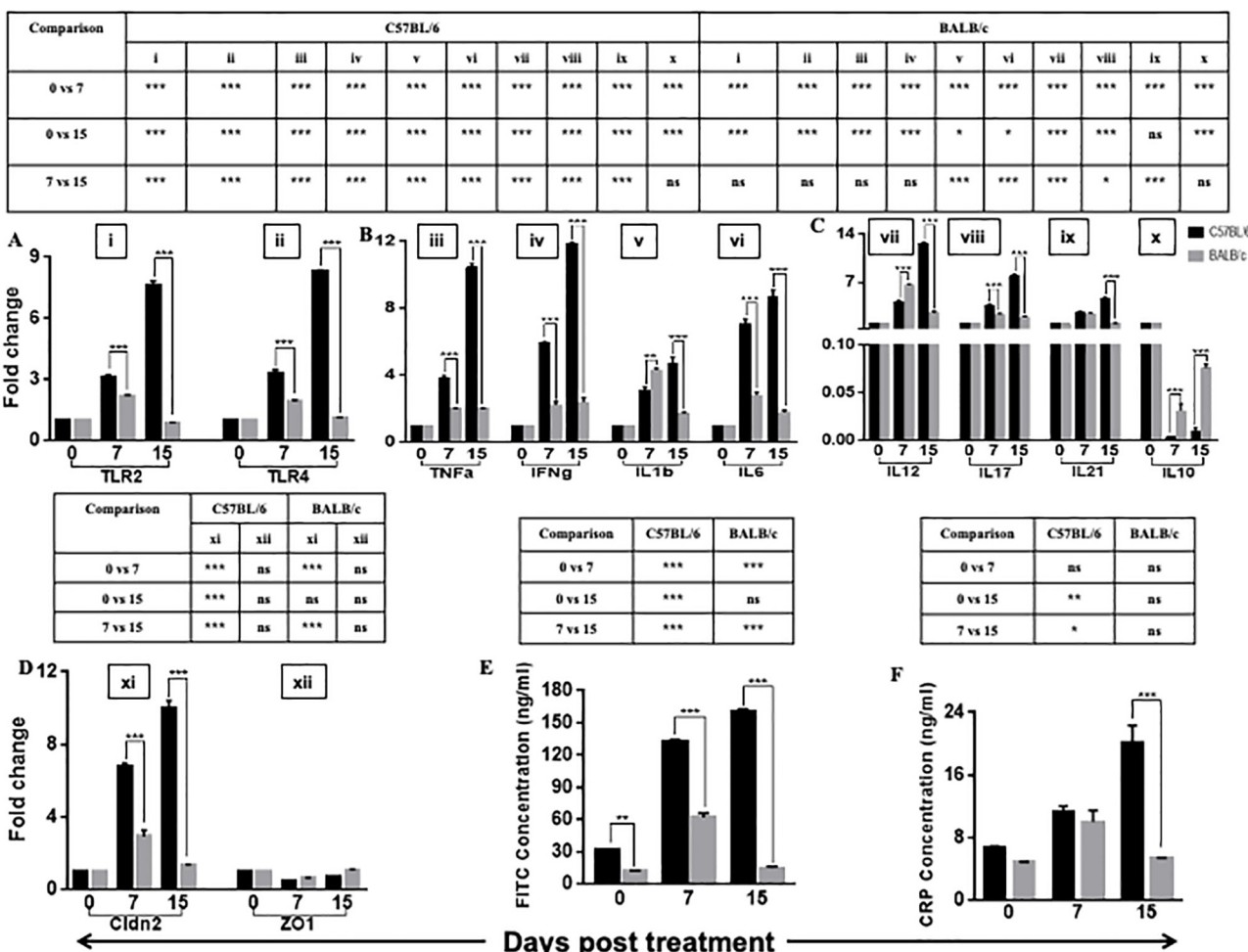

**Fig 1. Inflammatory responses in colon tissue following DSS treatment in C57BL/6 and BALB/c mice.** We represented the kinetics of different inflammatory responses, e.g., transcriptional expression of (A) Toll-like receptors (*TLR2* and *TLR4*) (B) pro (*TNFα*, *IFNγ*, *IL6*, *IL1β*), (C) (*IL12*, *IL21*, *IL17*) & anti-(*IL10*) inflammatory genes and (D) colonic tight junction related genes (*Cldn2*, *ZO1*) following DSS treatment in both C57BL/6 and BALB/c mice. We presented all values as means± SEM for 3 biological replicates. (E) We measured serum FITC-dextran levels in control and DSS-treated C57BL/6 and BALB/c mice to indicate intestinal permeability on days 0, 7, and 15 post-treatment as a consequence of inflammatory responses in the colon. (F) C- reactive protein in colon tissue of C57BL/6 and BALB/c mice was measured on days 0, 7, and 15 post-treatment to know the acute phase response of the host due to inflammation. The statistical significance among various days of treatment is denoted in the tables above respective panels to avoid unnecessary clutters. We used the roman numbering system above each gene name to keep the tables and figures tidy. We presented data as means± SEM (n = 6). We performed Two-way ANOVA followed by the Bonferroni test to determine the significance level. * corresponds to P<0.05, ** corresponds to P<0.01, *** corresponds to P<0.001.

DSS treatment (Fig 1B and 1C). We also measured the transcriptional change in *IL10* cytokine level as an activator of anti-inflammatory response. We observed significant downregulation of *IL10* (anti-inflammatory) and upregulation of pro-inflammatory genes in both treated mice strains till day 7 of the treatment condition. During the entire treatment period in C57BL/6, upregulated pro-inflammatory cytokines might be the reason for no significant upregulation of anti-inflammatory cytokine. On the contrary, anti-inflammatory cytokine level was significantly higher in the DSS-treated BALB/c group on day 15 than on day 7 of DSS treatment (Fig 1C). Expressions of all pro-inflammatory markers were substantially higher in C57BL/6 than BALB/c throughout the treatment condition, and anti-inflammatory gene expression was higher in BALB/c (Fig 1B and 1C).

Previous reports stated that uncontrollable immune reaction in the gut microenvironment leads to compromised gut barrier function [40]. The tight junction proteins, including occludin, claudins, and zonula occludens, play a crucial role in maintaining gut epithelial integrity [41]. As the effect of high inflammatory responses in the colon, we measured the gene expression of two tight junction proteins, i.e., *Cldn2* and *ZO1*. The previous reports found that the *Cldn2* gene is highly expressed in the gut tissue of colitis patients, whereas *ZO1*is downregulated. In the current study, the expression of *Cldn2* followed the same kinetics of inflammatory responses in both the mice strains. *Cldn2* expression was highest on day 15 of DSS treatment in C57BL/6 and day 7 of DSS treatment in BALB/c. Like the other inflammatory responses, *Cldn2* expression was significantly higher in C57BL/6 than BALB/c (Fig 1D). We did not find any significant changes in the gene expression of *ZO1* in either of the mice strains in any treatment condition (Fig 1D).

*Cldn2* gene is highly expressed in leaky gut epithelia [42]. We measured the serum FITC-Dextran level following DSS treatment to determine the gut leakiness. A high concentration of FITC-Dextran in the serum was an indication of high gut permeability. In C57BL/6, serum FITC-Dextran level increased gradually in the treated group compared to the control and reached its highest point on day 15 of DSS treatment. BALB/c gut permeability increased until day 7 following DSS treatment and gradually decreased as the DSS dose decreased (Fig 1E).

Activation of inflammatory cytokines, mainly *TNFα* and *IL6*, leads to the activation of the acute-phase response protein of the host [43, 44]. C- reactive protein (CRP) is one of such acute-phase response proteins. We measured the CRP level from colonic tissue and found that CRP level was significantly high only on day 15 of DSS treatment in C57BL/6 mice (Fig 1F).

## Colitis induced transcription and metabolic level changes of two differential immune bias host

To understand the molecular basis of the differential responses of two immune bias hosts in the presence of a common DSS dose, we performed transcriptomics analysis of colon tissue and metabolomics analysis of serum and cecal content of mice. This approach allowed us to get a clear idea about the role of host genetics and metabolic profile in differential disease responses in two different strains. We compared the transcriptomics and metabolite data from different disease severity levels for both strains of mice. Our results revealed differential responses of genetics (Fig 2A–2F) and metabolite profile (Fig 2G–2N), based on which we clustered them. First, we clustered the transcriptomics data, using LDA, based on the total no. of genes expressed in the colon at different treatment conditions in both strains (Fig 2A and 2B). Further, we shortlisted the genes based on the criteria mentioned in the methodology sections and again clustered them similarly for both strains of mice (Fig 2D and 2E). Mice from similar treatment conditions clustered together and clustered differentially based on different treatment conditions for both strains (Fig 2A, 2B, 2D and 2E). The shortlisted genes from the transcriptomics data are mainly related to the inflammatory responses of the host. The highest amount of, the highest In C57BL/6, genes were primarily associated with pro-inflammation, whereas in BALB/c, genes were related to anti-inflammation. We measured the distance between the clusters and the trajectory of the groups on the 2D plane to measure the extent of differential disease outcomes based on the immune background of the hosts [23, 24]. From the distance and trajectory of the groups on the 2D plane, we found in C57BL/6, the distance between control and treated groups increased gradually in both analyses, i.e., LDA of total genes (Fig 2C) and LDA of shortlisted genes (Fig 2F). This analysis revealed that the gene expressions of day 15 over day 0 were more different from day 7 over day 0 (S2 Table). On the contrary, in BALB/c, the distance between control and treated groups was almost similar in

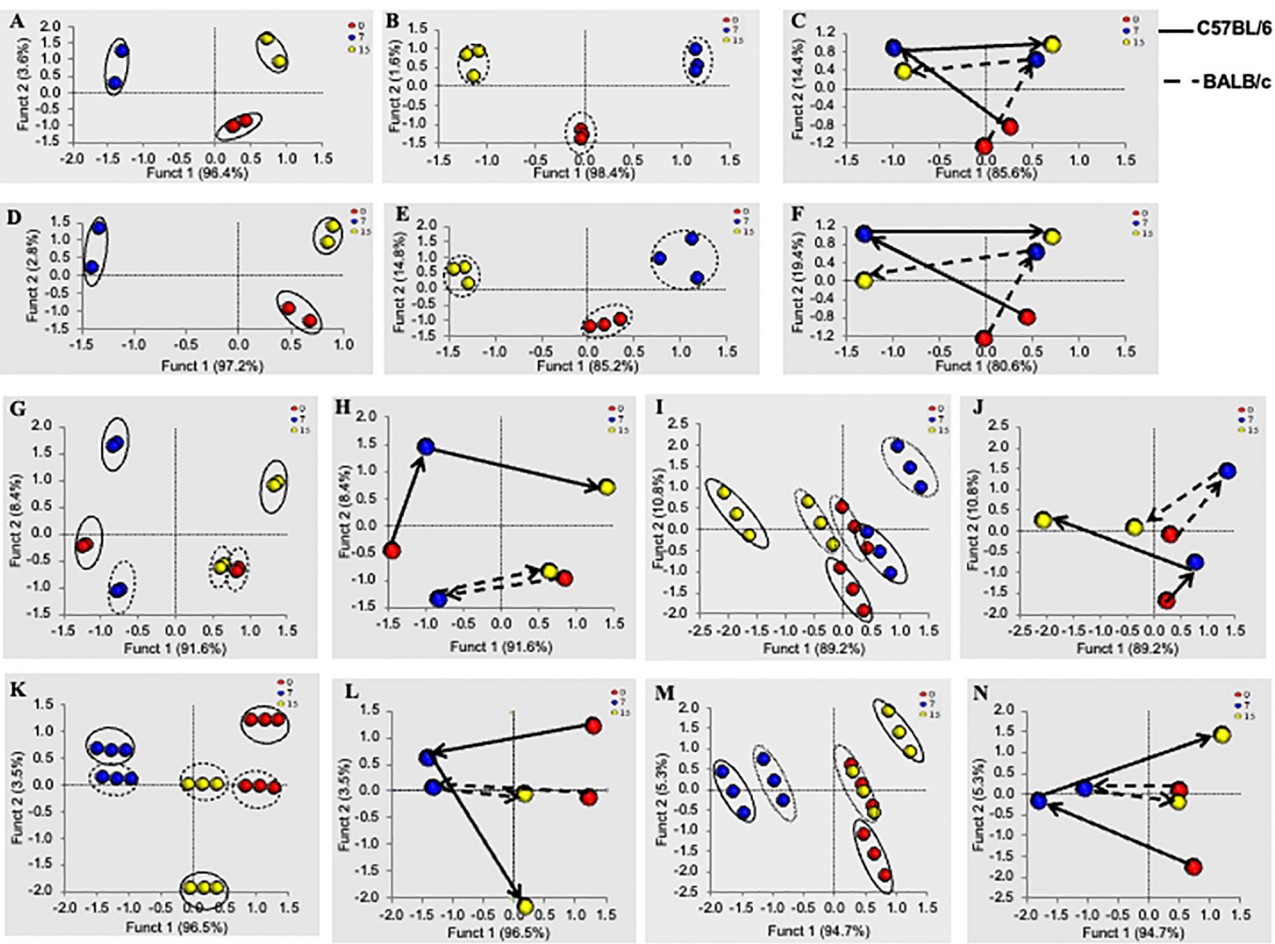

**Fig 2. Linear discriminant analysis of multi-omics data and the trajectory followed by different treatment conditions on the 2D plane.** We grouped the mice strain using Linear Discriminant Analysis (LDA) based on transcriptomics, metabolomics, and meta-metabolomics data according to the different treatment conditions. We showed LDA of the total gene expressed in the colon (A, B) and significantly altered genes in the colon (D, E) following DSS treatment on days 0, 7, and 15 in both C57BL/6 (A, D) and BALB/c (B, E) mice. We plotted the trajectory of LDA data of total gene expression (C) and significantly altered gene expression (F) in the colon to determine the differential responses based on different treatment conditions and different mice strains. Similarly, we showed LDA of total metabolites (G) and significantly altered metabolites (K) present in serum and total meta-metabolites (I) and significantly altered metabolites (M) present in cecal content followed by DSS treatment in both C57BL/6 and BALB/c mice. To know the differential responses based on the treatment conditions and different mice strains, we also plotted the trajectory of LDA data of total metabolites (H) and significantly altered metabolites (L) present in serum and total meta-metabolites (J) and significantly altered metabolites (N) present in cecal content. Distance between the clusters and the trajectory on the 2D plan for all the analyses is mentioned in detail in S2 Table. This analysis gives us a better understanding of differential responses of C57BL/6 and BALB/c in the presence of an equivalent dose of DSS treatment.

both analyses, i.e., LDA of total genes (Fig 2C) and LDA of shortlisted genes (Fig 2F). This analysis revealed that the gene expressions of day 15 over day 0 and day 7 over day 0 were almost similar without much change (S2 Table). This LDA analysis showed a similar kind of trend with the previously mentioned inflammatory responses of both strains.

Similarly, we performed LDA of metabolites (serum metabolites) and meta-metabolites (cecal metabolites) data. Like transcriptomics data, we performed LDA of all the altered metabolites and meta-metabolites as well as shortlisted metabolites and meta-metabolites. In both cases (LDA of all and shortlisted metabolites and meta-metabolites), mice from different treatment conditions clustered differentially on the 2D plane in C57BL/6 mice (Fig 2G and 2K). On the other hand, in BALB/c, mice from day 0 and day 15 clustered together and day 7

clustered differently on the 2D plane (Fig 2I and 2M). Like the transcriptomics data, we calculated the distance between different treatment conditions and trajectories on the 2D plane for metabolomics data to get an idea about the differential disease responses based on mice strains (Fig 2H, 2L, 2J and 2N). In C57BL/6, the distance between control and treated groups increased gradually in both analyses, i.e., LDA of total metabolites (Fig 2H) and meta-metabolites (Fig 2L) and LDA of shortlisted metabolites (Fig 2J) and meta-metabolites (Fig 2N). This analysis revealed that the metabolites and meta-metabolites concentration of day 15 over day 0 differed from day 7 over day 0 (S2 Table). On the contrary, in BALB/c, the distance between control and treated groups was almost similar in both analyses, i.e., LDA of total metabolites (Fig 2H) and meta-metabolites (Fig 2L) and LDA of shortlisted metabolites (Fig 2J) and meta-metabolites (Fig 2N). This analysis revealed that the metabolites and meta-metabolites concentration of day 15 over day 0 and day 7 over day 0 were almost similar without much change (S2 Table). Metabolomics analyses also followed the same trend of transcriptomics analyses and inflammatory responses of the hosts.

Varied distances between different clusters of transcriptomics and metabonomics data helped us understand the dissimilarity of genes, metabolites and meta-metabolites expression at other treatment conditions. We coined a new parameter, i.e., Dissimilarity Coefficient, to quantify the disparities in disease responses [23, 24]. We plotted the ratio of dissimilarity coefficient for different treatment conditions considering the mentioned parameters i)expressions of all altered genes and metabolites (Fig 3A) ii) significantly altered genes and metabolites (Fig 3B). We observed a distinct difference in the ratio of dissimilarity coefficient when we considered all altered genes and metabolites compared to the significantly altered genes and metabolites at different treatment conditions. Significantly altered parameters were thought to be the main driving force for differential disease outcomes in both strains of mice. To nullify the effect of non-significantly altered genes and metabolites (which may not have many roles in disease outcome) in the further analysis process, we considered only significantly altered genes and metabolites in the successive analysis process. In C57BL/6 mice, the highest difference or dissimilarities was found in significantly altered metabolite expressions at 15–0 over 0–7 (Fig 3B). This indicated that host metabolites played a critical role in the highest disease severity on day 15 of the DSS treatment. In BALB/c, meta-metabolites or microbiota-derived metabolites are crucial in determining the highest disease severity on day 7 of DSS treatment. The highest dissimilarity coefficient on 7–15 over 15–0 indicated a vast transition in meta-metabolite composition between day 7 and day 15 of the treatment condition (Fig 3B). To investigate the role of significantly altered genes and metabolites in disease severity, we calculated the disease severity index considering the factor altered genes and metabolites in different treatment conditions [25]. Unlike the dissimilarity coefficient, the role of host metabolites was highest in disease severity index in C57BL/6 on day 15–0 over 0–7 (Fig 3C). On the other hand, in BALB/c role of microbial metabolites (meta-metabolites) was highest in the disease severity index on days 7–15 over 15–0 (Fig 3C). Further analysis found that the altered genes and metabolites were either related to the activation of pro or anti-inflammatory processes of the host immune system. We calculated the ratio of pro- and anti-inflammatory genes, metabolites and meta-metabolites at different treatment conditions for both strains of mice. The ratio of Pro/Anti-inflammatory genes (Fig 3D) metabolites and meta-metabolites (Fig 3E) was highest on days 15–0 over 0–7 in C57BL/6. On the contrary, in BALB/c, the ratio was highest on days 7–15 over days 15–0 for genes (Fig 3D), as well as for metabolites and meta-metabolites (Fig 3E). We also detected the similarities of immunological responses between C57BL/6 and BALB/c in terms of genes. We found that the genes involved in inflammatory processes in C57BL/6 were different from BALB/c. Overall, the total number of genes involved in pro-inflammatory processes was much higher in C57BL/6 compared to BALB/c (Fig 3F). To investigate the

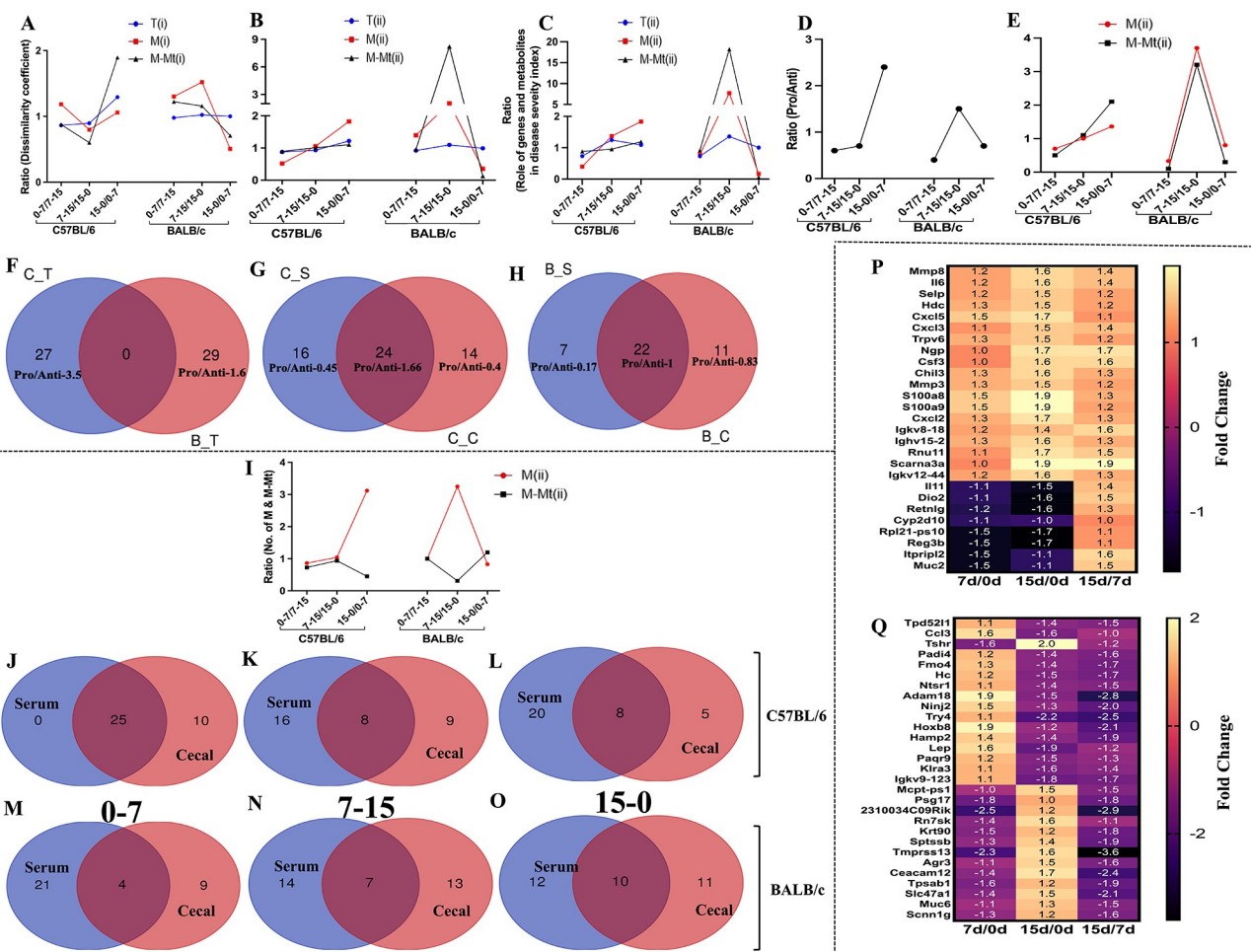

**Fig 3. Detailed analysis of altered transcriptomics and metabolomics status of the host at different stages of the disease.** From LDA, we have found that both mice strain formed three distinct clusters based on different treatment conditions. We have clustered the mice based on altered genes (T), metabolites (M), and meta-metabolites (M-Mt) expressions and measured the distances between different clusters on the 2D plane to quantify the difference between the treatment conditions. Every other bunch signifies a different treatment condition. We have measured the distance (dissimilarity coefficient) ratio of varying treatment conditions (clusters) to observe the kinetics of altered genes, metabolites, and meta-metabolites expressions at various treatment conditions for both the strains of mice. Panel (A) depicted the ratio of distances of different clusters when we considered all (i) the altered genes, metabolites, and meta-metabolites expressions for LDA analysis. Panel (B) depicted the ratio of distances of different clusters when we considered only the significantly (ii) altered genes, metabolites, and meta-metabolites expressions for LDA analysis. Further, we assess the influence of genetic and metabolic factors in determining the disease severity index (Panel C). We have quantified the ratio of pro and anti-inflammatory genes (Panel D) metabolites and meta-metabolites (Panel E) in different treatment conditions for C57BL/6 and BALB/c to know the role of inflammation in the determination of differential disease outcomes. We have also found that significantly altered genes were very much strain-specific (Panel F), and in C57BL/6 (C_T), the altered genes mainly were pro-inflammatory. In BALB/c (B_T), the altered genes were mostly related to anti-inflammation. To quantify the contributions of host metabolism and gut microbial metabolism in the inflammatory process, we calculated the ratio of significantly altered pro and anti-inflammatory metabolites from serum (S) and cecal content (C) of C57BL/6 (C) and BALB/c (B) mice. We found that in C57BL/6, host and microbial metabolites both triggered the inflammatory condition of the host (Panel G). On the contrary, in BALB/c, gut microbial metabolites contribute more to trigger the inflammatory condition of the host (Panel H). To investigate how the diversity of serum and cecal metabolites altered with altered inflammatory conditions, we calculated the ratio of no. of metabolites and meta-metabolites in different disease severity for both strains (Panel I). We also assessed how the no. of significantly altered common and unique metabolites from serum and cecal content changed along with the different treatment and disease severity levels in C57BL/6 (Panel J, K, L) BALB/c (Panel M, N, O) mice. The shortlisted genes from transcriptomics analysis responsible for differential disease progression in C57BL/6 (Panel P) and BALB/c (Panel Q) mice were depicted as the heat map in panels P and Q. Fold changes of the gene expressions were presented as a heatmap.

specific role of host and microbial metabolites (metabolites & meta-metabolites) in the inflammatory process, we determined the number of unique metabolites and meta-metabolites in C57BL/6 and BALB/c. It was found that both unique host and microbial metabolites contributed equally to the disease inflammatory process in C57BL/6 (Fig 3G). Whereas in BALB/c, the scenario was the opposite. Unique microbial metabolites took the main part in the activation of the inflammatory process of the host (Fig 3H).

Higher disease severity was the primary influencer of altered diversity (total no. of metabolites and meta-metabolites) of metabolites and meta-metabolites. We identified how the number of metabolites and meta-metabolites were changing with the varying disease severity. Increased disease severity increases the diversity of metabolites and decreases meta-metabolites' diversity in both C57BL/6 and BALB/c (Fig 3I). With different treatment conditions and severity levels, we also determined how the variety of common and unique serum and cecal metabolites changed in C57BL/6 and BALB/c. In C57BL/6, the diversity of unique serum metabolites increased with the increased disease severity, and the variety of unique cecal metabolites decreased (Fig 3J–3L). In BALB/c, with the increased disease severity diversity of unique serum metabolites decreased, and the diversity of unique cecal metabolites increased (Fig 3M–3O).

The shortlisted genes responsible for differential disease progression in C57BL/6 and BALB/c mice were depicted as a heatmap (Fig 3P and 3Q) and as tables for better understanding (S3A and S3B Table).

## Predicted altered metabolic and immune-related pathways due to the diseased condition of the host

We calculated the fold change of significantly altered metabolites (S4A and S4B Table) and meta-metabolites (S5A and S5B Table) and grouped them into two parts, i) upregulated and ii) downregulated. Using the KEGG pathway as a reference dataset, we predicted the altered metabolic pathways using these upregulated and downregulated metabolites and meta-metabolites. We named these pathways as upregulated and downregulated pathways of different treatment conditions. We represented the pathway names with a unique number. Pathway names with their corresponding unique pathway numbers were enlisted in the S6 Table. We plotted the bar graph with the topmost impacted pathways in various diseased conditions. The pathway analysis found that the pathways related to carbohydrate and nucleotide metabolisms were upregulated in the less severe condition of the disease and downregulated in the extreme form of the disease in both strains. In C57BL/6, pathways predicted from shortlisted metabolites (S3A and S3E Fig) and meta-metabolites (S3C and S3G Fig) related to carbohydrate and nucleotide metabolism were upregulated on day 7 over day 0 (S3A and S3C Fig) and downregulated on day 15 over day 0 and day 15 over day 7 (S3E and S3G Fig). On the other hand, in BALB/c, pathways predicted from metabolites were upregulated day 15 over day 0 and day 15 over day 7 (S3B Fig). For meta-metabolites, it was on day 15 over day 7 (S3D Fig). However, in BALB/c, we had not found any downregulated carbohydrate and nucleotide metabolism pathways from the pathways predicted from metabolites, but for meta-metabolites, it was on day 7 over day 0 (S3H Fig). Pathways related to amino acid and lipid metabolism were upregulated when the disease severity was highest in both mice strains and successively downregulated at the less severe colitis. In C57BL/6, the predicted pathways from metabolites (S3A and S3E Fig) and meta-metabolites (S3C and S3G Fig), related to amino acid and lipid metabolisms were upregulated on day 15 over day 0 and day 15 over day 7 (S3A and S3C Fig) and downregulated on day 7 over day 0 (S3E and S3G Fig). In BALB/c, the predicted pathways from metabolites (S3B and S3F Fig) and meta-metabolites (S3D and S3H Fig) related to the amino acid and

lipid metabolisms were upregulated on day 7 over day 0 (S3B and S3D Fig) and downregulated on day 15 over day 0 as well as day 15 over day 7 (S3F and S3H Fig). We performed a correlation analysis for C57BL/6 (S8A Table) and BALB/c (S8B Table) and found that carbohydrate and nucleotide metabolisms were positively correlated with anti-inflammation. In contrast, amino acid and lipid metabolism were positively correlated with pro-inflammation (S8A and S8B Table). This correlation analysis is well corroborated with our experimental observations.

We perform a joint pathway analysis to know the collective effect of gene-level and metabolite-level changes on the host system. This analysis considered both shortlisted genes and shortlisted metabolites and meta-metabolites to predict the biological pathways. Pathway names with their corresponding unique pathway numbers were enlisted in the S6 Table. Interestingly, we found that in C57BL/6, the highly impacted pathways from these two joint analyses, i.e., transcriptomics and metabolites (S3I Fig) and transcriptomics and meta-metabolites (S3K Fig), were mainly related to pro-inflammation related pathways along with the metabolic pathways. Pathways related to Toll-Like receptor, NOD- Like receptor, IL17, TNFα signaling, cytokine-cytokine receptor interactions were affected along with different metabolic pathways. Either one or more of these immune-related pathways were activated in all the treatment conditions, i.e., day 7 over day 0, day 15 over day 0, and day 15 over day 7. In BALB/c, the highly impacted pathways from these two joint analyses, i.e., transcriptomics and metabolites (S3J Fig) and transcriptomics and meta-metabolites (S3L Fig), no inflammation-related pathways were affected. Only metabolic pathways were involved in all the treatment conditions, i.e., day 7 over day 0, day 15 over day 0, and day 15 over day 7. This observation supports the point more strongly; the immune background of the host plays a crucial role in determining the disease severity of colitis.

## Altered gut microbial composition of diseased C57BL/6 and BALB/c

Transcriptional activation TLRs, other inflammatory cytokines, and compromised gut barrier function were the clear indication for diseased hosts' altered gut microbial composition. Activation of the host's inflammatory responses was the effect of more no. of gram-negative pathogenic bacteria in the gut. LPS (Lipopolysaccharide) secreted by these bacteria bind with the TLRs and activate the downstream pro-inflammatory pathways [45]. We investigated the gut microbial composition of treated groups and their time match controls at different diseased conditions for both strains. Data from our study revealed that the gut microbial composition of control C57BL/6 (Fig 4A) and BALB/c (Fig 4B) overtly belongs to the phyla Bacteroidetes and Firmicutes (Fig 4A and 4B). The abundance of the phyla Bacteroidetes (Fig 4C) and Firmicutes (Fig 4D) were reduced, while the abundance of Proteobacteria (gram-negative pathogens) phylum (Fig 4E) was increased significantly on day 7 and day 15 of DSS treated animals compared to their time-matched control. The proteobacteria level reached maximum by day 15 (32% of total abundance) in treated C57BL/6 and by day 7 (29% of total abundance) in treated BALB/c (Fig 4E). After day 7 of DSS-treated BALB/c showed a significantly different gut microbiota profile with the appearance of phylum, Verrucomicrobia, absent in treated C57BL/6 on day 15 (Fig 4F). A sudden increase in Bacteroidetes, Firmicutes, and Verrucomicrobia phyla, on day 15 of DSS-treated BALB/c replaced the predominance of Proteobacteria phylum (Fig 4F). This result again supports our previous observation about the differential responses of two different immune-bias mice in the presence of a common DSS dose.

Since each phylum contains various genera, it is crucial to know the significant changes in the abundance and diversity at the genus level following DSS treatment in C57BL/6 (Fig 4G) and BALB/c (Fig 4H). At the genus level, the gut microbiota of untreated time-matched control of either type of mice majorly composed of genus Bacteroides and Alisteps of

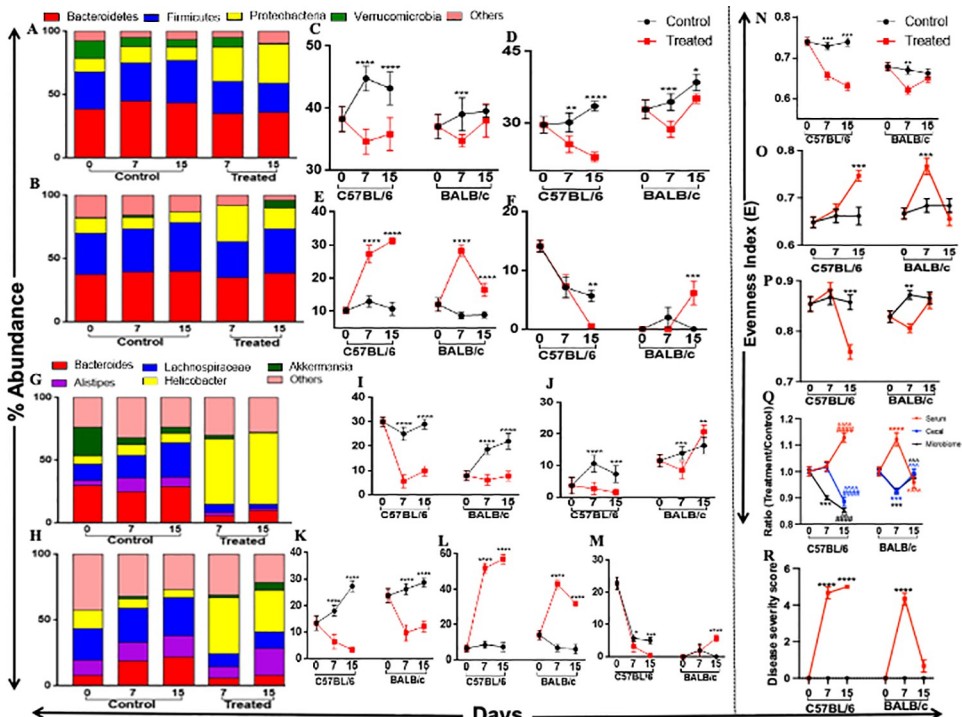

**Fig 4. Phylum and genus level changes in gut microbial composition and altered metabolic and microbial diversity due to altered disease severity.** Stack bar showing the relative changes in significant phyla of gut microbiota in control and DSS treated C57BL/6 (A) and BALB/c (B) mice. We have plotted the abundance of significant phyla of control and its respective treated group for both the mice strains for better understanding. (C, D, E, F) depicts the following phylum respectively- Bacteroidetes, Firmicutes, Proteobacteria, Verrucomicrobia. Panel (H, I) represents the relative changes in the genus level of gut microbiota in control, and DSS treated C57BL/6 (H) and BALB/c (I) mice. Similarly, like phylum, we have plotted the abundance of the different genus of control and its respective treated group for both the mice strains for better understanding. (I, J, K, L, M) depicts the following genus respectively- Bacteroides, Alisteps, Lachnospiraceae, Helicobacter, Akkermensia. (N) Further, we calculated the changes in gut microbial diversity. We represented the kinetics of phylum-level diversity by calculating the Equitability (Evenness) index (E) of gut microbiota for both control and treated groups of mice for both the strains. Altered gut microbial composition and diversity could be the main reason for the host's altered metabolic profile and diversity. (O, P) We represented the kinetics of metabolite (O) and meta-metabolite diversity (P) by calculating the equitability index of metabolite and mete-metabolite composition for both control and treated groups C57BL/6 and BALB/c mice. (Q) We also calculated the (treatment/control) ratio of microbial and metabolic Evenness index (E) of C57BL/6 and BALB/c mice to get a comprehensive idea of microbial and metabolic changes based on disease severity. (R) Disease severity score was calculated to get an idea of how altered microbial and metabolic profiles affect disease severity followed by DSS treatment. We presented all values as means± SEM for 3 biological replicates. We performed Two-way ANOVA followed by the Bonferroni test to determine the significance level. */#/^ corresponds to P<0.05, ** /##/^^corresponds to P<0.01, ***/###/^^^ corresponds to P<0.001, ****/####/^^^^ corresponds to P<0.0001. (In panel Q, * symbolizes the comparison between day 0 and day 7, # symbolizes the comparison between day 0 and day 15, ^ symbolizes the comparison between day 7 and day 15).

Bacteroidetes phylum and genus Lachnospiraceae from Firmicutes phylum. In the DSS treated condition, the abundance of Bacteroides (Fig 4I), Alisteps (Fig 4J), and Lachnospiraceae (Fig 4K) significantly decreased and predominated by the genus Helicobacter (Fig 4L) from Proteobacteria phylum in both the strains. The abundance of Helicobacter genus was highest on day 15 of DSS treatment in C57BL/6 and day 7 in BALB/c (Fig 4L). In BALB/c, similar to the phylum level, the genus belongs to the Bacteroidetes (genus-Alisteps), Firmicutes (genus- Lachnospiraceae), and Verrucomicrobia (genus- Akkermensia) phyla replace the predominancy of genus Helicobacter of Proteobacteria phylum (Fig 4J–4M). In C57BL/6, no such changes were observed on day 15 of DSS treatment.

The above results revealed that along with the disease severity, a load of pathogenic Proteobacteria in the gut significantly increased till the end of the DSS treatment in C57BL/6. In DSS-treated BALB/c, as the disease severity decreased after day 7 of DSS treatment, the abundance of Proteobacteria also significantly reduced in the gut.

In addition, we further determined the changes in the diversity (the diversity and evenness in the distribution of microbial and metabolite composition) of gut microbial composition and metabolites and meta-metabolites diversity along with the altered disease severity [19]. The diversity Evenness index at the microbial phylum level showed a significant gradual decrease in diversity throughout the treatment condition in C57BL/6 compared to their time-matched control. On the contrary, in BALB/c, on day 7, we observed a substantial reduction in the diversity in the treatment condition compared to their time-matched control (Fig 4N). Diversity Evenness index of serum and cecal metabolites behaved oppositely. Serum metabolite diversity increased with the disease severity (Fig 4R) in both strains compared to their time-matched control (Fig 4O). On the contrary, cecal metabolite diversity decreased with the disease severity (Fig 4R) in both strains compared to their time-matched control (Fig 4P). The ratio of the Evenness index of the treatment/control condition also supports the mentioned observation in both strains (Fig 4Q). The disease severity score was calculated using the criteria mentioned in the S7 Table.

## Microbiota regulated metabolic pathways of diseased C57BL/6 and BALB/c

Reports from previous studies suggested that the specific phylum is responsible for the metabolism of particular nutrients. Not all phyla can metabolize all kinds of nutrients, e.g., Phyla i) Bacteroidetes and Verrucomicrobia are responsible for carbohydrate metabolism ii) Firmicutes is responsible for nucleotide metabolism iii) Proteobacteria is responsible for amino acid and lipid metabolism [46–48].

In this current study, we observed a positive correlation between the i) abundance of Bacteroidetes and Verrucomicrobia and carbohydrate metabolism ii) abundance of Firmicutes with nucleotide metabolism iii) abundance of Proteobacteria with amino acid and lipid metabolism in both the strains (S8A and S8B Table). We also found a positive correlation between anti-inflammation, the abundance of Bacteroidetes, Firmicutes, and carbohydrate and nucleotide metabolism. On the other hand, pro-inflammation, the abundance of Proteobacteria and amino acid, and lipid metabolism were positively correlated (S8A and S8B Table). We already mentioned in previous sections that as the colitis disease severity increased, carbohydrate and nucleotide metabolism became downregulated and amino acid and lipid metabolism upregulated. This section plotted the already predicted metabolic pathways acquired from the metabolites and meta-metabolites data, with the responsible phyla for the upregulation or downregulation of these pathways.

As the abundance of Bacteroidetes (S4C and S4G Fig) and Verrucomicrobia (S4D and S4H Fig) decreased with the disease severity, carbohydrate metabolism also downregulated in C57BL/6 (S4A and S4B Fig, S4A- pathways predicted from serum metabolites, S4B- pathways predicted from cecal metabolites) and BALB/c (S4E and S4F Fig, S4E- pathways predicted from serum metabolites, S4F- pathways predicted from cecal metabolites) mice. The same trend was followed by nucleotide metabolism. Nucleotide metabolism (S4I, S4J, S4L and S4M) and abundance of Firmicutes (S4K and S4N Fig) phylum became downregulated with disease severity in C57BL/6 (S4I and S4J Fig, S4I- pathways predicted from serum metabolites, S4J-pathways predicted from cecal metabolites) and BALB/c (S4L and S4M Fig, S4L- pathways predicted from serum metabolites, S4M- pathways predicted from cecal metabolites) mice. On the contrary, as the disease severity increased, the abundance of Proteobacteria increased (S4Q

and S4T Fig). This further leads to the upregulation of amino acid and lipid metabolism pathways in C57BL/6 (S4O and S4P Fig, S4O- pathways predicted from serum metabolites, S4P-pathways predicted from cecal metabolites) and BALB/c (S4R and S4S Fig, S4R- pathways predicted from serum metabolites, S4S- pathways predicted from cecal metabolites) mice.

### The probable mechanism of different metabolic conversion and the associated gut microbial genus at the different inflammatory states of the host

So the overall observations of all the experiments gave us a crystal clear idea that host-immune background is the primary regulatory factor in determining colitis disease severity. Host immune bias is crucial in activating inflammatory responses, altering the host's metabolic and gut microbial composition in the diseased condition. Increased amino acid metabolism in inflammatory conditions leads to a high amount of lactate and glutamate production. We observed a high amount of butyrate production from its precursor molecules in the control or recovery condition. The gut microbial genus responsible for butyrate production was predominant at the disease's control and recovery phase. The probable mechanism of the metabolic interconversions at particular inflammatory conditions is depicted in Fig 5.

## Discussion

We comprehensively examined the severity responses of DSS induced colitis in two immunologically bias mice strains. With a special mention, we are perhaps the first group that used a composite DSS dosage (5% for the 1st week+ 2.5% for the 2nd week) to understand all the stages of colitis in terms of severity in two different immune-biased mice models within a brief period, i.e., 2 weeks. Data from the gene-based inflammatory study and multi-omics approach revealed that in the presence of a higher dosage (5% for the 1st week), the extent of colonic inflammation and other metabolic and gut microbial changes were almost similar for both strains of mice. At the lower continuing dosage (2.5% for the 2nd week) of DSS, the inflammatory condition was more pronounced and severe in C57BL/6, whereas BALB/c started recovering from the inflammatory state and reaching the normal healthy condition.

Like human colitis, DSS caused damages to intestinal epithelial cells and activated the host's inflammatory responses [49]. After colitis induction in both mice strains, damage-associated molecular markers were released by the damaged colon tissue bound with the Toll-Like Receptors (mainly Tlr2 and Tlr4) and activated flares of inflammatory responses first-line defense mechanism of hosts. Activation of TLRs further activates monocytes and macrophages by using the nucleotide-binding oligomerization domain 2 protein (NOD2) pathway [36, 37]. Activation of monocytes and macrophages produced an array of soluble pro-inflammatory cytokines, e.g., TNF-α, IFN-γ [38, 50, 51]. IL-12 and IL-21 also mediate IFN-γ secretion due to Th1 mediated pro-inflammatory response [52, 53]. TNF-α exerts its pro-inflammatory effects through increased production of IL-1β and IL-6 [54]. IL-1β and IL-6 produced by damaged and inflamed tissue stimulate the production of another inflammatory cytokine, i.e., IL17 [55].

The present study revealed that *TLR2* and *TLR4* genes activated significant pro-inflammatory responses in both mice strains on the 1st week of DSS treatment. We observed a significant upregulation of all the pro-inflammatory mediators (*TNF-α, IFN-γ, IL6, IL1β, IL12, IL21, IL17*). C57BL/6 followed the trend till the end of the treatment. In BALB/c, no further upregulation was observed in the gene expression of pro-inflammatory mediators on the 2nd week. In some cases, the expression of pro-inflammatory genes tends to reach their basal level. To know the reason for the amelioration of colitis severity in BALB/c, we checked the expression of one of the most crucial anti-inflammatory mediators, IL10. CD4+ Th2 cells trigger the secretion of

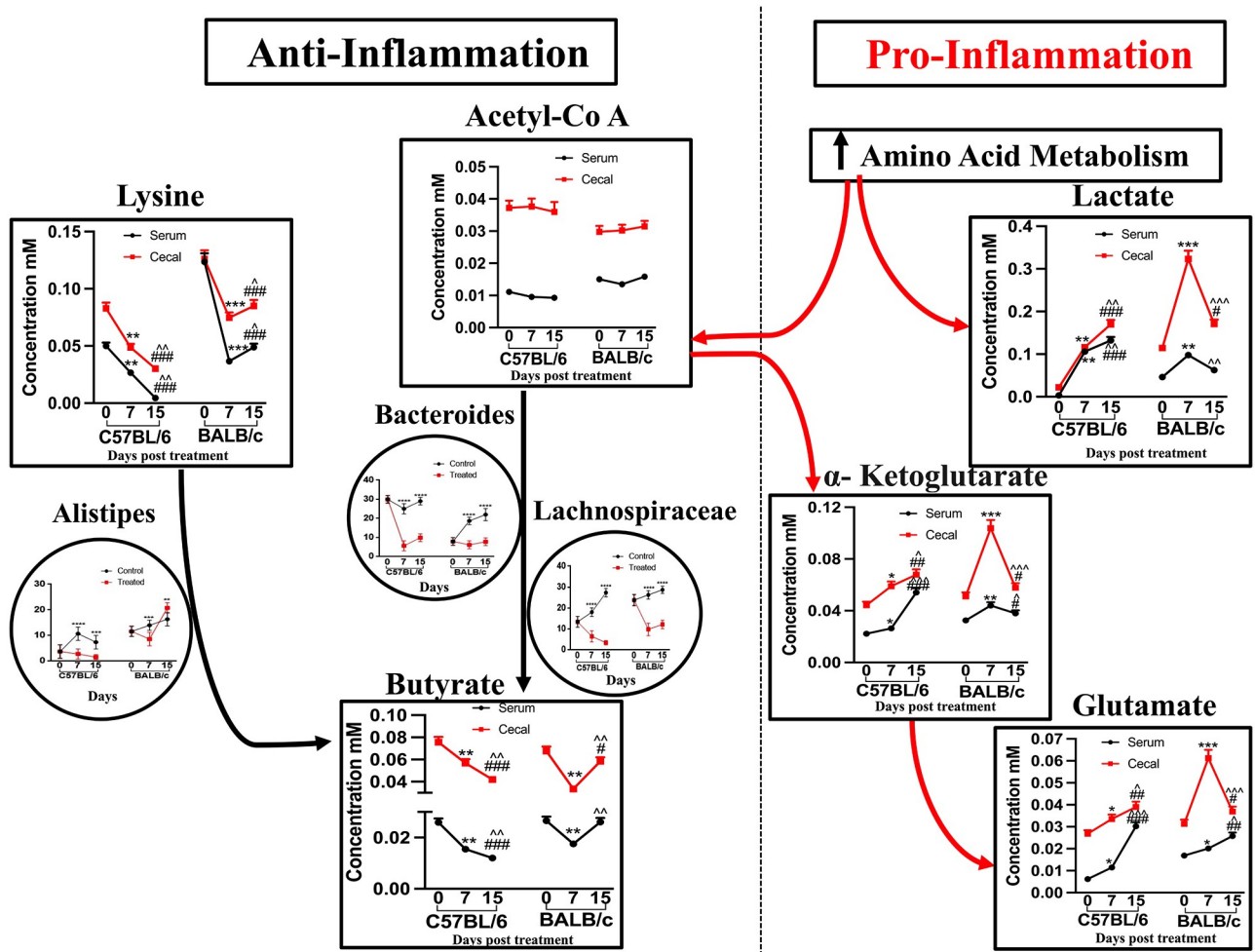

**Fig 5. A predictive schema of different regulatory metabolic pathways and associated microbial genus contributing to the varied inflammatory condition at different severity levels of the disease in C57BL/6 and BALB/c mice.** We performed Two-way ANOVA followed by the Bonferroni test to determine the significance level. */#/^ corresponds to P<0.05, ** /##/^^corresponds to P<0.01, ***/###/^^^ corresponds to P<0.001, ****/####/^^^^ corresponds to P<0.0001. * symbolizes the comparison between day 0 and day 7, # symbolizes the comparison between day 0 and day 15, ^ symbolizes the comparison between day 7 and day 15.

IL-10 as an anti-inflammatory response of the host [56]. As we expected, the *IL10* gene expression was upregulated in BALB/c as the disease severity decreased. No such upregulation of *IL10* expression was observed in C57BL/6 in any of the treatment conditions.

Activation of the intestinal pro-inflammatory responses is the critical factor for the breakdown of intestinal tissue homeostasis. Severe inflammatory reactions in intestinal epithelium ultimately lead to leaky gut formation [40, 57]. A transmembrane protein known as the Tight Junction (TJ) proteins regulates gut permeability in healthy conditions. TJ proteins maintain gut permeability by forming heteropolymer strands at the apical pole of the basolateral membrane of gut epithelia [41, 58]. Cldn2 and ZO1 are two TJ proteins involved in colitis disease pathology. The *Cldn2* gene is highly expressed in the gut tissue of colitis patients, whereas *ZO1* is downregulated [40–42, 59]. The current study revealed that gut barrier function compromised with an upregulation of *Cldn2* gene expression in colon tissue as the disease severity increased. No change was observed in *ZO1* gene expression in either of the mouse strains. The

*Cldn2* gene is highly expressed in leaky gut epithelia [42]—the leaky gut condition corroborated with the *Cldn2* expression level.

Activation of the pro-inflammatory pathway has many other consequences in the host system, such as activating the host acute phase response protein [43, 44, 51]. In the current study, we observed significant upregulation of CRP in C57BL/6 on day 15 of the treatment but not in any treatment conditions in BALB/c mice.

To dig deeper, how the host immune background reacts to the diseased condition differently, we evaluated systemic level changes of the host. Transcriptomics and the untargeted metabonomics approach helped us assess the systemic changes in two different mice strains in diseased conditions. LDA analysis of colon transcriptomics data and metabonomics study from serum and cecal content deciphered the change in individual animals of both strains and further clustered them based on the differential disease outcome. Varied trajectory and Dissimilarity Coefficient (distance) between different clusters on the 2D plane revealed the extent of differential responses, in gene and metabolite level, between different severity levels in two different mice strains [23, 24]. Increased Dissimilarity Coefficient between different clusters indicated more intense differential respond between different treatment conditions. With increased dissimilarity between clusters, both the strains' disease severity index was also increased [25]. The genes, metabolites, and meta-metabolites involved in the higher disease severity index were mainly related to the activation of the pro-inflammatory responses in the host. The genes that were involved in pro-inflammation were intensely host-specific. The Th1 immune background C57BL/6 was probably the main reason for activating more pro-inflammatory genes than BALB/c throughout the diseased condition. The role of the host and microbial metabolites in the disease process was also quite different in C57BL/6 and BALB/c. In C57BL/6, the unique host and microbial metabolites both contributed equally to activating the host's inflammatory responses. In contrast the unique microbial metabolites were the primary influencer in activating host inflammatory responses in BALB/c mice. Varied disease severity was hypothesized to be the main reason for altered host and microbial metabolic diversity. Increased disease severity might be the main reason for increased diversity of serum/host metabolites and decreased diversity of cecal/microbial metabolites in both strains of mice. However, the number of unique serum metabolites were increased with the disease severity in C57BL/6 and decreased in BALB/c. In the case of cecal metabolites, the total opposite trend was found. The number of unique serum metabolites was reduced with the disease severity in C57BL/6 and increased in BALB/c. From the mentioned observation, this can be concluded that in C57BL/6 altered number of unique host or serum metabolites had a more significant influence on disease severity. In contrast, a modified number of unique microbial or cecal metabolites had a more significant impact on the disease severity of BALB/c. So, in a nutshell, data from LDA analysis revealed that disease severity responses were highly different from each other on days 0, 7, and 15 of DSS treatment in C57BL/6. In BALB/c day 0 and 15 resembles the same level of severity compared to day 7 of DSS treatment.

The shortlisted genes, metabolites, and meta-metabolites used in LDA analysis were further used to predict multiple biological pathways responsible for exerting differential disease outcomes in the host. Due to the altered metabolic condition of the host, Altered biological pathways were categorized into two parts. Amino acid and lipid metabolism pathways were upregulated as the severity of the disease increased in both strains. Upregulation of carbohydrate and nucleotide metabolism was thought to be one of the main reasons for easing the diseased condition in BALB/c. Observation from the earlier study prompted increased CD4+ cell population, and IL22 cytokine expression in colitis patients was tightly correlated with upregulation of carbohydrate and nucleotide metabolism and downregulation of amino acid and lipid metabolism. High CD4+ and IL22 expressions collectively create potent anti-

inflammatory reactions to ameliorate the inflammatory state of the disease [60–66]. As the inflammatory condition was more concentrated in C57BL/6, it showed a continuous upregulation in amino acid and lipid metabolism throughout the treatment period. Lesser disease load was strong enough to activate the carbohydrate and nucleotide metabolism at the less severe form of the disease in BALB/c. Extremely severe inflammatory reactions in C57BL/6 were potent enough to activate other inflammation-related biological pathways. The genetic and metabolic joint pathway prediction revealed that along with the metabolic pathways, pathways related to inflammatory reactions, e.g., Toll-Like receptor, NOD- Like receptor, IL17, TNFα signaling, cytokine-cytokine receptor interactions were upregulated throughout the treatment conditions. No such changes were observed in BLAB/c. From the observation, this can be concluded that non-significant differences in the gene level at the diseased condition compared to control might be the reason for protecting the diseased state. Altered metabolic functions were thought to be a factor for the modified gut microbial composition of the host, followed by DSS treatment. Previous reports suggested that the host's upregulated amino acid and lipid metabolism is the consequence of the high Proteobacterial load in the gut. LPS secreted by this phylum activates the downstream inflammatory responses [66–68]. A kinetics study of the gut microbial composition revealed that as the inflammation and amino acid and lipid metabolism increase with time, Proteobacterial abundance, mainly the abundance of genus Helicobacter, increased in the gut content C57BL/6. In BALB/c, we observed a surge in the Proteobacterial level in the gut on day 7 of DSS treatment. On day 15 of DSS treatment, Proteobacteria abundance was suppressed by 3 other phyla, i.e., Bacteriodetes, Verrucomicrobia, and Firmicutes. It is already well established that Bacteroidetes and Verrucomicrobia are responsible for increased carbohydrate metabolism, and Firmicutes are responsible for nucleotide metabolism [47, 48, 69]. A high abundance of Bacteriodetes, Verrucomicrobia, and Firmicutes, activated carbohydrate and nucleotide metabolism, which further triggered the anti-inflammatory responses in BALB/c.

Varying disease severity, the altered gut microbial composition was the probable reason for the increased abundance of one particular microbial phylum or a group of host or microbial metabolites, which would repress the abundance of other phyla or metabolites. As a sequela, the overall evenness index of the gut microbiota and metabolites would change [19, 25]. In the current study, we observed an increase in Proteobacteria phylum with the disease severity associated with a decrease in gut microbiota diversity. Gut microbial evenness index was significantly decreased in C57BL/6. On the contrary, in BALB/c, the gut microbial evenness index was significantly reduced on day 7 and restored on day 15 of the DSS treatment. This might be another possible reason for less severe disease outcomes in BALB/c.

In the case of the metabolic evenness index, the microbial metabolite evenness index followed the same trend. In contrast, the opposite trend was observed in the case of the host metabolite evenness index. From here, this can be concluded that the host and its gut microbial counterpart contributed separately in differential disease severity of both strains of mice.

Increased amino acid metabolism at the highest inflammatory condition of the disease in both strains of mice prompted us to investigate the downstream mechanism of change in the host and microbial metabolic profile and their role in disease severity. A probable mechanism of varied disease severity is clearly explained in the study. The severe inflammatory condition leads to acidosis by producing a high amount of lactate and glutamate from acetyl-CoA [70–74]. In the control and recovery state of the disease, the production of the high amount of short-chain fatty acid, butyrate, from lysine and acetyl-CoA was the essential metabolic product to maintain the anti-inflammatory condition of the host [75–77]. The notable genus responsible for the interconversion of butyrate from lysine and acetyl-CoA were Alistipes, Bacteroides, and Lachnospiraceae [78, 79]. Increased abundance of Alistipes, Bacteroides, and

Lachnospiraceae on day 15 of the treatment condition led to a high amount of butyrate and activated the anti-inflammatory responses in BALB/c. No such notable change in beneficial microbial genus and metabolites ultimately caused a prolonged disease severity in BALB/c.

## Conclusion

The present study could conclude that the oral administration of a specific (5% for one week followed by 2.5% for the 2nd week) DSS dose could produce reproducible colitis in C57BL/6 BALB/c mice. DSS-induced colitis activated the Th1 immune responses in the host system. Thus the immunological background of C57BL/6 mice might have directed to exhibit tremendous severity response. We noticed a time-dependent increase in inflammation and disease severity after lowering the DSS dosage. Th2 immunological background of BALB/c, on the contrary, perhaps helped the mice to maintain a low disease severity compared to C57BL/6.

Observation from a multi-omics study prompted that the Th1-skewed immune background of C57BL/6 altered the host's overall homeostasis by changing the host's genetic, metabolic, and microbial composition. Altered genes, metabolites, and gut microbiota collectively created a niche for highly severe inflammatory reactions flares. The tolerogenic nature of BALB/c could control the inflammation to regain its normal homeostasis very easily. Activation of anti-inflammatory responses protected BALB/c from the activation of long-term inflammatory reactions, which further control its genetic, metabolic, and gut-microbial composition.

More interestingly, in C57BL/6, the disease severity was mainly hosted factor-driven. Activation of more pro-inflammatory genes and host metabolites was the main reason for altered disease severity. Gut microbiota and its associated metabolites had a less critical role in disease severity due to less activation of pro-inflammatory microbial metabolites. On the contrary, altered gut microbial composition and associated microbial changes had more stake in determining disease severity in BALB/c. Activation of more pro-inflammatory microbial metabolites than host metabolites and genes is the probable reason for this hypothesis.

Although C57BL/6 and BALB/c exerted different severity responses in a common DSS dose, the pattern of metabolic and gut microbial changes was similar in both strains at the highest severity of the disease. Inflammation leads to the high amino acid and lipid metabolism, leading to the high abundance of Helicobacter genus under Proteobacteria phylum in the host's gut. These parameters all together create a niche for the severe form of colitis in the host. The amino acid and lipid metabolism pathways and the mentioned genus Helicobacter could be a promising and efficient biomarker in the characterization of colitis irrespective of the immune-bias condition of the host. On the other hand a high abundance of short-chain fatty acid butyrate and Alistipes, Bacteroides and Lachnospiraceae genus could be used as a potential biomarker to determine the recovery phase of the disease. These biomarkers might also help understand the disease etiology better and diagnosing the disease early.

## Supporting information

**S1 Fig. Altered clinical score, colon length, and histopathological changes of both mice strain followed by DSS treatment.** (A) Physiological changes of both mice strains were used to determine the clinical score to measure the severity of the disease progression. (B, C) Altered colon length (B) and histopathological scoring (C) further showed the disease progression in both mice strains. We presented all values as means± SEM for 3 biological replicates. We performed Two-way ANOVA followed by the Bonferroni test to determine the significance level. *corresponds to $P<0.05$, ** corresponds to $P<0.01$, *** corresponds to $P<0.001$. (TIFF)

**S2 Fig. A summary slide depicted the significant findings of the current model system.**
(TIFF)

**S3 Fig. Majorly impacted pathways due to the altered metabolic and inflammatory responses in the presence of DSS treatment.** (A, B) Majorly impacted pathways due to altered metabolites' upregulation at various treatment conditions in serum in C57BL/6 (A) and BALB/c (B) mice. (C, D) Majorly impacted pathways due to altered metabolites' upregulation at various treatment conditions in cecal content in C57BL/6 (C) and BALB/c (D) mice. (E, F) Majorly impacted pathways due to altered metabolites' downregulation at various treatment conditions in serum in C57BL/6 (E) and BALB/c (F) mice. (G, H) Majorly impacted pathways due to altered metabolites' downregulation at various treatment conditions in cecal content in C57BL/6 (G) and BALB/c (H) mice. (I, J) Primarily impacted pathways due to the alteration of serum metabolites and changes in the host's inflammatory responses in both C57BL/6 (I) and BALB/c (J) mice. (K, L) Primarily impacted pathways due to the alteration of cecal metabolites (meta-metabolites) and changes in the host's inflammatory responses in both C57BL/6 (K) and BALB/c (L) mice. We represented the pathway names with numbers to avoid unnecessary clutters. We have enlisted the pathway names with their corresponding pathway numbers in S6 Table.
(TIFF)

**S4 Fig. Majorly impacted metabolic pathways in different treatment conditions and the corresponding bacterial phyla responsible for the metabolic changes.** We have categorized the metabolic pathways into four different groups- i) Carbohydrate metabolism, ii) Nucleotide metabolism, iii) Amino acid metabolism, and iv) Lipid metabolism. Bacteroidetes and Verrucomicrobia are responsible for affected Carbohydrate metabolism. Firmicutes are responsible for affected Nucleotide metabolism. Proteobacteria are responsible for affected Amino acid and lipid metabolism. In C57BL/6 mice, (A, B) represents the affected carbohydrate metabolism in serum (A) and cecal (B) level, (I, J) represents the affected nucleotide metabolism in serum (I) and cecal (J) level and (O, P) represents the affected amino acid and lipid metabolism in serum (O) and cecal (P) level. (C, D, K, Q) represents the phylum level abundance of Bacteroidetes, Verrucomicrobia, Firmicutes, and Proteobacteria, respectively, in gut microbiota along with its corresponding metabolic process. In BALB/c mice, (E, F) represents the affected carbohydrate metabolism in serum (E) and cecal (F) level, (L, M) represents the affected nucleotide metabolism in serum (L) and cecal (M) level and (R, S) represents the affected amino acid and lipid metabolism in serum (R) and cecal (S) level. (G, H, N, T) represents the phylum level abundance of Bacteroidetes, Verrucomicrobia, Firmicutes, and Proteobacteria, respectively, in gut microbiota along with its corresponding metabolic process. We represented the pathway names with numbers to avoid unnecessary clutters. We have enlisted the pathway names with their corresponding pathway numbers in S6 Table.
(TIFF)

**S1 Table. Sequences of forward (F) and reverse (R) primers used in gene expression studies.**
(DOCX)

**S2 Table. Quantitative estimation of differential responses of DSS treated C57BL/6 and BALB/c mice in terms of transcriptomics, metabolomics and meta-metabolomics using Linear Discriminant Analysis (LDA).**
(DOCX)

**S3 Table.** Significantly affected genes from transcriptomics study with function and fold change values at different treatment conditions for A. C57BL/6 and B. BALB/c mice. (DOCX)

**S4 Table.** Significantly affected serum metabolites from metabonomics study with their function and fold change values at different treatment conditions for A. C57BL/6 and B. BALB/c mice. (DOCX)

**S5 Table.** Significantly affected cecal metabolites from metabonomics study with their function and fold change values at different treatment conditions for A. C57BL/6 and B. BALB/c mice. (DOCX)

**S6 Table. Pathway name and its corresponding pathway no. used in the main figure panel.** (DOCX)

**S7 Table. Disease severity score based on stool texture and rectal bleeding.** (DOCX)

**S8 Table.** Correlation analysis between inflammatory parameters, microbial abundance (phylum level) and metabolic pathways in A. C57BL/6 and B. BALB/c mice. (DOCX)

## Acknowledgments

We are thankful to the NISER animal house facility for assistance in preparing the slides for histopathological analysis. We also acknowledge the help of Mr Krushna Chandra Murmu, a graduate student at the Institute of Life Sciences, Bhubaneswar, in preliminary transcriptomic data calling and sharing.

## Author Contributions

**Conceptualization:** Sohini Mukhopadhyay, Palok Aich.

**Data curation:** Subha Saha, Punit Prasad, Palok Aich.

**Formal analysis:** Sohini Mukhopadhyay, Palok Aich.

**Funding acquisition:** Palok Aich.

**Methodology:** Sohini Mukhopadhyay, Subhayan Chakraborty, Arindam Ghosh, Palok Aich.

**Project administration:** Palok Aich.

**Resources:** Palok Aich.

**Supervision:** Palok Aich.

**Writing – original draft:** Sohini Mukhopadhyay.

**Writing – review & editing:** Palok Aich.

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
