## [Decision Letter · Decision Letter 0]

1 Nov 2021

PONE-D-21-31676Differential colitis susceptibility of Th1- and Th2-biased mice- A Multi-Omics ApproachPLOS ONE

Dear Dr. Aich,

Thank you for submitting your manuscript to PLOS ONE. After careful consideration, we feel that it has merit but does not fully meet PLOS ONE’s publication criteria as it currently stands. Therefore, we invite you to submit a revised version of the manuscript that addresses the points raised during the review process.

We look forward to receiving your revised manuscript.

Kind regards,

Lili Chen

Academic Editor

PLOS ONE

Journal Requirements:

 [The funders had no role in study design, data collection and analysis, decision to publish, or preparation of the manuscript.]

Reviewers' comments:

Reviewer's Responses to Questions

**Comments to the Author**

1. Is the manuscript technically sound, and do the data support the conclusions?

Reviewer #1: Yes

Reviewer #2: Yes

2. Has the statistical analysis been performed appropriately and rigorously? 

Reviewer #1: Yes

Reviewer #2: Yes

3. Have the authors made all data underlying the findings in their manuscript fully available?

Reviewer #1: Yes

Reviewer #2: Yes

4. Is the manuscript presented in an intelligible fashion and written in standard English?

Reviewer #1: Yes

Reviewer #2: Yes

5. Review Comments to the Author

Reviewer #1: Mukhopadhyay et al performed experiments and analyses aimed at addressing the potential mechanisms underlying the differential colitis susceptibility between C57BL/6 (type 1-prone) and BALB/c (type 2-prone) mice. They employed multi-omics approaches including comparative transcriptomics, metabonomics, and metagenomics analyses. They found that, in C57BL/6, the severity of DSS-induced colitis was mainly host factor-driven, including pro-inflammatory host genes and host metabolites. On the contrary, in BALB/c, the disease severity was more affected by the microbiota, such as gut microbial composition and microbial metabolites. These are very useful datasets which could be very helpful for other investigators in choosing mouse strains to study DSS-colitis or other disease models, as well as in considering the candidate pro-/anti-inflammatory host genes, host metabolites, microbial strains, or microbial metabolites reported in this manuscript as potential mechanisms.

I only have some minor comments:

1. Lines 210 to 213 - “The shortlisted genes from the transcriptomics data are mainly related to the inflammatory responses of the host. In C57BL/6, genes were primarily associated with pro-inflammation, whereas in BALB/c, genes were related to anti-inflammation.” Given that these conclusion are foundations of this manuscript, I would suggest the authors to use a heat map to show the shortlisted genes as a part of a main figure.

2. Tables S3, S4 and S5 were only mentioned in the Materials and Methods section which is a waste of good data. I would suggest the authors to cite these tables in their main text.

Reviewer #2: The manuscritps presents a thorough analysis of DSS induced colitis in Th1 and Th2 biased mice strains, C57BL/6 and BALB/c resectively. In order to find clinical differences between strains, they use first a high dose of DSS (1 week), followed by a lower dose (1 week). En the second week of treatment BALB/c mice recover, while C57BL/6 don´t. Surprisingly, this results are not shown, although they could go to a suplementary figure. No data on clinical scores, hystology or bowel weight/length is presented, making results weaker. A simple figure describing the findings in this particular model should be included.

Than the authors study cytokines (Pro-inflamatory and tolerogenic) at different stages. Results regarding cytokines are quite consisting with other papers. The multiomics approach, regarding bowel genes and metabolites compares to changes in gut microbiota are a unique and highly interesting contribution of the article. The data analysis are sound and rigorous statiscal test are performed. The English is quite clear and easy to read.

The results are well dicussed and enrich the article.

6. PLOS authors have the option to publish the peer review history of their article (what does this mean?). If published, this will include your full peer review and any attached files.

Reviewer #1: No

Reviewer #2: No

---

## [Author Response · Author response to Decision Letter 0]

28 Dec 2021

Reviewers' comments:

Reviewer's Responses to Questions

Comments to the Author

1. Is the manuscript technically sound, and do the data support the conclusions?

Reviewer #1: Yes

Reviewer #2: Yes

2. Has the statistical analysis been performed appropriately and rigorously?

Reviewer #1: Yes

Reviewer #2: Yes

3. Have the authors made all data underlying the findings in their manuscript fully available?

Reviewer #1: Yes

Reviewer #2: Yes

4. Is the manuscript presented in an intelligible fashion and written in standard English?

Reviewer #1: Yes

Reviewer #2: Yes

5. Review Comments to the Author

Reviewer #1: Mukhopadhyay et al performed experiments and analyses aimed at addressing the potential mechanisms underlying the differential colitis susceptibility between C57BL/6 (type 1-prone) and BALB/c (type 2-prone) mice. They employed multi-omics approaches including comparative transcriptomics, metabonomics, and metagenomics analyses. They found that, in C57BL/6, the severity of DSS-induced colitis was mainly host factor-driven, including pro-inflammatory host genes and host metabolites. On the contrary, in BALB/c, the disease severity was more affected by the microbiota, such as gut microbial composition and microbial metabolites. These are very useful datasets which could be very helpful for other investigators in choosing mouse strains to study DSS-colitis or other disease models, as well as in considering the candidate pro-/anti-inflammatory host genes, host metabolites, microbial strains, or microbial metabolites reported in this manuscript as potential mechanisms.

I only have some minor comments:

1. Lines 210 to 213 - “The shortlisted genes from the transcriptomics data are mainly related to the inflammatory responses of the host. In C57BL/6, genes were primarily associated with pro-inflammation, whereas in BALB/c, genes were related to anti-inflammation.” Given that these conclusion are foundations of this manuscript, I would suggest the authors to use a heat map to show the shortlisted genes as a part of a main figure.

Answer: We appreciate the efforts of the reviewers. We tried to incorporate the valuable comments by the reviewers. As reviewer #1 suggested to use a heatmap of shortlisted genes as the main figure for a better understanding of the data. We have followed the instructions and incorporated the heatmaps in the main figure (Figure 3P and 3Q ) and also explained in detail in the main text of the manuscript (Line No. 603-605) and also in figure legends (Line no. 638-640).

2. Tables S3, S4 and S5 were only mentioned in the Materials and Methods section which is a waste of good data. I would suggest the authors to cite these tables in their main text.

Answer: We have incorporated and cited all the three tables in the main text as per the reviewer’s suggestion. Table S3 (Line No. 605), Table S4 (Line No. 644), Table S5 (Line No. 645) all three tables were well explained in the main text. 

Reviewer #2: The manuscritps presents a thorough analysis of DSS induced colitis in Th1 and Th2 biased mice strains, C57BL/6 and BALB/c resectively. In order to find clinical differences between strains, they use first a high dose of DSS (1 week), followed by a lower dose (1 week). En the second week of treatment BALB/c mice recover, while C57BL/6 don´t. Surprisingly, this results are not shown, although they could go to a suplementary figure. No data on clinical scores, hystology or bowel weight/length is presented, making results weaker. A simple figure describing the findings in this particular model should be included.

Answer: We thank the reviewer #2 for this comment. We have incorporated the data of clinical and histopathological scores and bowel length of the animal at control and DSS treated conditions. We have provided the data as supplementary figure (S1_Fig) and also mentioned in the main text in the appropriate places (Line No. 385-392). As reviewer suggested to provide a simple figure describing the model in a better way. We have provided the major and unique findings of the current model as supplementary figure (S2_Fig) and mentioned in the manuscript's main text (Line No. 393-394). 

Than the authors study cytokines (Pro-inflamatory and tolerogenic) at different stages. Results regarding cytokines are quite consisting with other papers. The multiomics approach, regarding bowel genes and metabolites compares to changes in gut microbiota are a unique and highly interesting contribution of the article. The data analysis are sound and rigorous statiscal test are performed. The English is quite clear and easy to read.

The results are well discussed and enrich the article.

---

## [Editor Report · Decision Letter 1]

10 Feb 2022

Differential colitis susceptibility of Th1- and Th2-biased mice- A Multi-Omics Approach

PONE-D-21-31676R1

Dear Dr. Aich,

We’re pleased to inform you that your manuscript has been judged scientifically suitable for publication and will be formally accepted for publication once it meets all outstanding technical requirements.

Kind regards,

Lili Chen

Academic Editor

PLOS ONE

---

## [Editor Report · Acceptance letter]

16 Feb 2022

PONE-D-21-31676R1 

Differential colitis susceptibility of Th1- and Th2-biased mice: a multi-omics approach 

Dear Dr. Aich:

I'm pleased to inform you that your manuscript has been deemed suitable for publication in PLOS ONE. Congratulations! Your manuscript is now with our production department. 

Kind regards, 

on behalf of

Dr. Lili Chen 

Academic Editor

PLOS ONE